# Psychosocial Impact of False-Positive Newborn Screening Results: A Scoping Review

**DOI:** 10.3390/children11050507

**Published:** 2024-04-24

**Authors:** Jane Chudleigh, Pru Holder

**Affiliations:** Cicely Saunders Institute, King’s College London, London SE5 9PJ, UK; pru.holder@kcl.ac.uk

**Keywords:** newborn screening, psychological, psychosocial, false positive, anxiety, depression

## Abstract

Psychosocial consequences of false-positive results following newborn bloodspot screening have been identified as a potential risk to this highly successful public health initiative. A scoping review was undertaken in October 2023 underpinned by the Arksey and O’Malley framework. Twenty-four papers were included in the review, many of which focused on cystic fibrosis. The results indicated that impact of false-positive results is variable; some studies suggest false-positive results have the potential to result in negative sequelae including increased stress and changes in parental perceptions of their child, while others suggest these impacts are transient and, in some instances, may even lead to positive outcomes. Further evidence is needed to ensure the representation of other conditions included in newborn bloodspot screening and to support strategies to overcome potential negative sequela.

## 1. Introduction

Newborn bloodspot screening (NBS) is a highly successful public health initiative. It allows for presymptomatic identification and early initiation of treatment for babies affected by genetic or congenital conditions, leading to better health outcomes for the child [1]. However, while the clinical and fiscal benefits of NBS are undisputed, the potential psychological impact on the child’s family, particularly following a false-positive result, needs careful consideration.

The number of conditions included in NBS differs between countries (e.g., the United States (US): 63; the United Kingdom (UK): 9; and Australia: 27 [2]). In addition, the number of conditions included in NBS since its inception in the 1960s has continued to rise and with it the number of false-positive results [3]. False-positive rates differ depending on both the condition screened for and the approach used. In the UK, the positive predictive value (the number of children with a positive NBS result that have the condition) ranges from 50% (inherited metabolic conditions including maple syrup urine disease; isovaleric academia; glutaric aciduria type 1; and homocystinuria) to 95% (sickle cell disorder) [1]. The introduction of genomic NBS has the potential to further increase the number of false-positive results depending on whether (i) a sensitive or specific approach is adopted, as a sensitive approach would reduce the positive predictive value but ensure affected individuals are not missed and (ii) reporting is not restricted to affected individuals with pathogenic or likely pathogenic variants [4].

Informing parents of their child’s positive NBS result may challenge parental perceptions of health, illness and disease for several reasons [5,6]. Parents usually seek medical assistance for their child if they are unwell or are displaying symptoms suggestive of an underlying health condition. However, the intention of NBS is to identify a potential condition in an otherwise asymptomatic child. For most parents, a positive NBS result follows a healthy pregnancy including ultrasound scans that do not raise concerns, and an unremarkable delivery and newborn physical examination [7]. In addition, parents have reported that information given at the time of NBS did not aways facilitate the informed consent process due to a lack of perceived choice [8,9,10]. Consent practices differ between countries, with some countries operating an ‘opt in’ approach (e.g., the UK) and others an ‘opt out’ approach (e.g., most of the US and Canada). Parents’ experiences of information giving and their ability to make informed choices can also be limited by the information received about the NBS process and the conditions that are being screened for. Parents have reported receiving false reassurance at the time of NBS, that NBS is ‘routine’ or ‘always comes back negative’ [7,10], which can lead to parents being ill prepared to receive a positive NBS result. Parents are also often unaware of their own genetic status and therefore receiving a positive NBS for their child also has the potential to uncover unsolicited genetic information about themselves. Being informed that their child does not have the condition following diagnostic testing after a positive NBS result is a relief for parents, but false-positive NBS results can complicate parents’ perceptions of their child in addition to the trustworthiness of the process. This paper discusses condition-specific as well as generic consequences for the child and family, following false-positive NBS results.

## 2. Materials and Methods

A scoping review was undertaken in October 2023 and updated in March 2024 using the first five stages of the Arksey and O’Malley framework [11]: (1) identifying the research question; (2) identifying relevant studies; (3) study selection; (4) charting the data; and (5) collating, summarizing, and reporting the results. The review followed the Preferred Reporting Items for Systematic reviews and Meta-Analyses (PRISMA) and its extension for Scoping Reviews (PRISMA-ScR) [12]. MEDLINE (Ovid interface) and APA PsycInfo were searched from inception to 2024 to answer the following question: ‘What are the psychosocial impacts of false-positive results following NBS?’ Search terms were ((newborn screening.mp. OR exp Neonatal Screening/) AND (false positive.mp.) AND (psychosocial.mp. OR psychological.mp. OR anxi*.mp. OR distress*.mp. OR uncertain*.mp.)). Reference lists of eligible articles and relevant papers were screened to identify additional papers not found via the database search.

Inclusion criteria were primary research studies or systematic reviews that reported psychosocial outcomes following a false-positive result for any condition included in NBS. Exclusion criteria were studies that focused on other screening programmes (e.g., newborn hearing screening, newborn physical examination), books, editorials, commentary or opinion pieces or conference abstracts and papers published in a language other than English as translation was beyond the scope of this review.

The Covidence platform (https://www.covidence.org/, accessed 12 April 2024) was used to screen the results of the search and perform data extraction. Two reviewers (JC and PH) screened the titles and abstracts and full texts; no disagreements emerged. JC and PH also performed data extraction using the Covidence data extraction tool 1. Data extraction focused on the population, concept and context (PCC) framework [13] and included the author, year and country where the study was conducted; the condition(s) studied; the data collection methods employed; the sample; and outcomes in terms of the psychosocial impact of false-positive NBS results. Narrative synthesis was used to synthesize the findings of the included studies [14]. An inductive approach was used to organize and summarize the results from the included papers to answer the research question. The results from included papers were read several times to identify patterns, similarities and differences in terms of the psychosocial impacts of false-positive results following NBS.

## 3. Results

The database search yielded 53 studies; no additional studies were identified via the reference list search. Following screening, 24 studies were included [15,16,17,18,19,20,21,22,23,24,25,26,27,28,29,30,31,32,33,34,35,36,37,38] in the scoping review (See Figure 1). Most focused on the impact of false-positive NBS results for cystic fibrosis [17,18,19,20,25,29,30,31,34,36] and metabolic/endocrine disorders [15,16,21,22,26,27,33,37]. An overview of the included studies can be seen in Table 1. Findings are presented under the following themes: reactions to the initial NBS result; emotional reactions to false-positive NBS results and impact of false-positive NBS results on reproductive decision making; parental perceptions of child vulnerability following false-positive NBS results; and healthcare utilization following false-positive NBS results.

### 3.1. Reactions to the Initial NBS Result

Studies focused on the time between NBS and diagnostic testing and/or the period between the NBS result and any repeat testing that may have been deemed necessary. An early study with parents of children who had an initial positive NBS result for one of the metabolic conditions found that parents who were aware that the initial NBS result was abnormal were no more anxious or depressed while waiting for the repeat test results than other parents. In addition, parents were less anxious and depressed after the repeat test was found to be ‘normal’ [37]. However, a later review found that abnormal NBS results, even when repeat testing was normal (i.e., a false-positive result), were associated with parental anxiety and/or depression [28]. An early study in the US found parents expressed shock, disbelief and confusion in response to the initial positive NBS result [34]. A study in the UK with mothers following a false-positive NBS result for CF found that nearly two thirds recalled that the wait for the results of repeat tests was difficult and led to emotions such as upset, guilt and increased anxiety [18]. This was supported by a study conducted in Canada which found distress associated with notification of the initial positive NBS result for CF and it being described as ‘the scariest time of [their] lives’ [17]. Also, a study in the US which included false-positive results for a range of conditions described this period as a ‘nightmarish two and a half weeks’ [23]. Finally, in a study in the US focused on false-positive NBS for Krabbe disease, all parents disclosed they experienced emotional distress and negative emotions, specifically uncertainty during the initial notification and leading up to the follow-up appointment with a genetic counsellor. In addition, many reported feelings of emotional distress, including increased anxiety and uncertainty during their appointment with the genetic counsellor, but felt reassured by the information provided [33]. These mirror findings of studies with parents of children who have received true-positive results following NBS [7,39,40,41]. This is understandable since parents of children who received both false- and true-positive results would have similar experiences in terms of being informed of their child’s NBS result and undergoing diagnostic testing [1]. However, the issue is that for those who ultimately end up with a false-positive result, this experience is unnecessary as their child does not need, and will therefore not benefit from, an early identification and initiation of treatment. In addition, a false-positive result has the potential to lead to longer term negative psychosocial sequalae.

### 3.2. Emotional Reactions to False-Positive NBS Results

Diagnostic testing provides confirmation of a true- or false-positive result. Studies have reported mixed findings about whether the negative emotions associated with an initial positive NBS result subside over time or not following a confirmed false-positive result.

For instance, positive outcomes were noted with parents in the US who had received a false-positive result following CF NBS, believing that it had strengthened their relationship, brought them closer and offered them a new perspective on their lives [20]. Similarly, another study in the US, which included families who had received a false-positive result for various conditions, found the experience had helped them set priorities, gain perspective and focus on important things. Parents also reported appreciation and gratitude for the experience [21] and an appreciation of the importance of NBS [33].

An early US study with parents who received a false-positive result for CF found most (92%) were relieved following the negative sweat test and (88%) were aware that a negative sweat test meant their child did not have CF. Those who were contacted by telephone were more likely to misunderstand the sweat test results compared to those told in person [34]. A French study found that Perceived Stress Scale scores decreased and were comparable to the French population 3 months after a false-positive NBS result for CF [19]. This was supported by a Canadian study exploring the impact of false-positive NBS for CF using validated scales to measure anxiety, distress, maternal perception of child vulnerability and perceived uncertainty related to childhood illness. Findings indicated that two months after the child’s birth and one year later, mean anxiety and distress scores were low and did not differ from a control group [17]. Another study which explored anxiety six months after false-positive NBS for CF found that scores from the Hospital Anxiety and Depression Scale (HADS) did not differ significantly between parents who had received a false-positive NBS results and a control group. In addition, parents in the false-positive group who were well informed about NBS (measured using a knowledge questionnaire), were less likely to experience anxiety and depression (*p* < 0.018) [36]. Furthermore, a recent study exploring parents’ experiences of NBS for CF found that parents of children with a false-positive NBS result reported less negative feelings following confirmatory diagnostics than those of children with CF. However, 17% remained anxious about the result [30]. Another more recent Canadian study explored the psychological impact of expanded NBS (specifically in relation to NBS for an inborn error of metabolism, an endocrine disorder (congenital adrenal hyperplasia and congenital hypothyroidism) or cystic fibrosis) using the Parenting Stress Index and the Depression, Anxiety and Stress Scale. Scores indicated no significant differences between mothers who had received true-negative, true-positive and false-positive results for their child in the previous four–six months. Condition-specific data were not reported [16].

This contrasts with findings of a very recent study that compared outcomes for parents who had received true-positive, false-positive or inconclusive NBS results with parents who had received negative (normal) NBS results for all screened conditions in the Netherlands. Findings indicated that parents who had received true-positive and false-positive NBS results reported more negative emotions compared to controls four months after NBS (*p* < 0.001) [35]. In addition, two US studies found that six months after receiving NBS results for metabolic disorders, mothers in the false-positive group scored significantly higher (*p* < 0.001) on the Parenting Stress Index and the parent–child dysfunction subscale (*p* < 0.001) compared to mothers whose children had received a negative NBS result [15,21]. Similarly, a study in the US found that almost 10% of parents who had received a false-positive NBS result for a metabolic or endocrine disorder for their child reported clinically significant stress [22]. While one of these studies also found fathers in the false-positive group scored significantly higher (*p* < 0.001) on the Parenting Stress Index [15], the other did not support this [21]. Finally, a systematic review concluded that anxiety in false positives may not return to normal levels; some residual anxiety may remain, possibly over extended periods of time [32].

### 3.3. Impact of False-Positive NBS Results on Reproductive Decision Making

Receiving a false-positive NBS result could also impact future reproductive decision making. An early study in the US found that, following a false-positive NBS result for CF, 69% did not change their reproductive plans as a result of the false-positive result, 8% definitely changed their plans and 22% were uncertain [34]. In a more recent study, a minority of parents who had received a false-positive NBS result for CF in the US stated that the experience had changed their minds (8%) or were uncertain about having additional children in the future [29]. Parents in another study conducted in the US reported guilt related to being a carrier of CF and having passed the faulty gene to their child [20]. Conversely, a study in the US with parents whose children had received a negative sweat test following a positive CF NBS result found fewer than half of untested couples expressed an interest in genetic testing, suggesting these parents were less concerned about the false-positive NBS result in relation to their future reproductive decision making [31]. A study in the US found that parents of children who had received a false-positive NBS result for a metabolic or endocrine disorder were more likely to express a desire for not having more children, whereas parents of children who had received a true-positive NBS result were more likely to emphasize a fear of recurrence of the disorder in future pregnancies [22]. A Canadian study found that following a false-positive result for CF, maternal worry scores were associated with concerns around future reproductive decision making for their child who was a CF carrier [17].

### 3.4. Parental Perceptions of Child Vulnerability following False-Positive NBS Results

A study conducted in the US with parents of children who had received a positive NBS result for CF but were later identified as carriers found the experience had given the parents an increased appreciation of their child’s good health [20]. A French study found that Vulnerable Child Scale scores three months following a false-positive CF NBS result were associated with low parental perception of child vulnerability [19]. Similarly, a study in Canada found that vulnerability scores post false-positive CF NBS results were low compared to reference means [17]. Also, a study conducted in the Netherlands found that there were no significant differences in parental perceptions of child vulnerability or parent-reported child health status for those who had received true-positive, false-positive, inconclusive and negative (normal) NBS results for all conditions included in screening 4 months after receiving the result [35]. This is in contrast to a study with parents who had received a false-positive NBS result for one of the metabolic conditions, which found that 36% of parents reported concern about the health of their infant because of the need for the repeat test [37].

An early study conducted in the US to explore the psychological impact of false-positive results of NBS for CF found that following a false-positive NBS result, many parents expressed lingering concerns that their child might still have CF or need a repeat diagnostic test in the future [29]. Another study conducted in the US with parents of children who had received a positive NBS result for CF but were later identified as carriers reported being concerned that their child’s gene could transform over time to cause illness. Parents also expressed mistrust about the accuracy of the NBS test. For some parents, this led them to be hypervigilant for signs of respiratory problems in their child. The false-positive result could also lead to concerns about other family members, including worries their older children could be an unknown carrier as well as other family members who had historically had respiratory symptoms [20]. It should be noted that this study was conducted prior to the global harmonization process to provide a consistent international approach to the management of children with an uncertain outcome following CF NBS, called CF screen-positive inconclusive diagnosis (CFSPID). This is important since a proportion of children with a CFSPID designation are at risk of converting to a CF diagnosis or developing a CFTR-related disorder [42,43]. This could therefore explain parents’ uncertainty at the time the study was undertaken, and also their concerns about their child becoming symptomatic at a later stage. Conversely, a study in the US which explored parental understanding following a negative sweat test after an initial positive NBS result for CF found that 94% of parents understood their child did not have CF after a negative sweat test and 79% of those identified as a carrier understood their result [31]. Another study in the US with parents who had received a false-positive result for Krabbe disease found a few parents reported ongoing anxiety and concern related to a misunderstanding regarding the possible risk for late-onset Krabbe disease and carrier status. Therefore, concerns around later manifestations of disease following a false-positive result may not be limited to CF [33].

### 3.5. Healthcare Utilization following False-Positive NBS Results

A study conducted in the US found no difference in healthcare utilization by the age of six months between parents of children who had received a false-positive NBS result for a metabolic condition and parents who had received a negative NBS result [24]. This is in contrast to an earlier review which found an increase in emergency room visits and hospitalization of infants following a false-positive NBS result [28]. In addition, a study in the US which found that ≥six months after false-positive results for metabolic disorders, twice as many children had been hospitalized compared to a comparison group of children who had received a negative NBS result [21]. This is supported by a Canadian study which found that a greater proportion of children with a false-positive result following CF NBS had more than two outpatient visits and more than two hospital admissions when compared with children who had received a negative NBS result [26]. Similarly, a US study found that pre-term but not term infants with false-positive NBS results had more acute outpatients visits than their counterparts with normal NBS results [38]. Another Canadian study also found higher rates of physician visits, emergency department visits, and inpatient hospitalizations following a false-positive NBS result for medium-chain acyl-CoA dehydrogenase deficiency compared to children who had received a negative NBS result [27]. Similarly, a study in China found that more mothers of children who had received a false-positive result for a metabolic disorder reported their child required extra parental care and that they were worried about the child’s future development compared with mothers of children who had received a negative NBS result. In addition, children who had received a false-positive result were three times as likely to experience hospitalization [26]. Finally, a recent study conducted in the Netherlands which included all conditions included in NBS found that 5 weeks and 4 months after NBS, parents who had received true-positive and false-positive NBS results reported significantly more visits to their paediatrician when compared with parents who had received negative (normal) NBS results. Interestingly, in the same study, parents who had received a false-positive NBS result reported more frequent hospital admissions 5 weeks after NBS compared to parents who had received negative (normal) NBS results, but this was no longer observed 4 months after NBS, indicating their concerns may have resolved [35].

## 4. Discussion

While the studies included in this review have identified the potential for both immediate and longer term sequalae associated with a false-positive result, both these and the wider literature have also identified strategies to reduce the impact of false-positive results on families. These highlight the importance of recognizing NBS as a journey rather than simply the point at which the NBS sample is taken or the timeframe around the initial result/diagnostic outcome delivery [44].

Antenatally, this includes providing parents with a clear explanation of NBS and all potential outcomes as well as informing them about the likelihood of a false-positive NBS result if these data are available [16,22,36]. This is important since the clinical spectrum in children who have received a positive NBS result varies widely and therefore the information provided to parents needs to be carefully constructed to prepare them adequately for a range of outcomes following diagnostic testing. A quantitative stated preference study found parents are able to and would like to be involved in how they received information antenatally as part of the NBS programme. This study suggested a need to use different communication approaches depending on the parent group being targeted that may be dependent on parent characteristics or the type of NBS condition [45]

When the NBS sample is taken, a discussion with parents focused on any preferences they have for receiving the results would provide the opportunity for the person taking the sample to openly discuss the possibility of a positive NBS result and avoid false reassurance [10,22,23,26,46,47,48]. Potential impacts on parental relationships may be mitigated by ensuring the mother is not alone when the NBS result is communicated [7] so they are not responsible for communicating the result to other family members. In addition, information should be tailored when delivering a positive NBS result so that the information related to the signs and symptoms of the suspected condition is only shared if this could lead to complications before the child is seen for diagnostic testing. The purpose of this would be to avoid parents becoming hypervigilant while waiting for any repeat tests and/or following the false-positive result [10,18,41]. Communication of positive NBS results is a subtle and skilful task which demands thought, preparation and evidence to minimize potentially harmful negative sequelae [7,40,49,50]. Guidance regarding the content and most appropriate approach to communication with parents may be variable, depend on the condition being screened for, and may not be evidence-based [1,51]. Previous studies have highlighted the importance of the approach used (face-to-face vs. telephone or letter), the knowledge and experience of the person imparting the positive NBS result, as well as the content of the message [52,53,54]. Parents and health professionals often report a preference for face-to-face communication as opposed to a telephone call. The reasons for this are multifaceted and include providing a more personable approach, being able to judge the emotional response of the recipient and tailor information accordingly, as well as issues related to a lack of consideration regarding where the parent might be and who might be with them if telephone communication is used [10].

Increased stress and anxiety associated with a false-positive NBS result were more commonly reported in earlier studies [15,21,22,28,29]. This contrasts with some of the more recent studies which reported similar anxiety and stress levels in parents who had received false-positive NBS results compared to parents who had received negative NBS results [16,17,30]. This is promising and suggests that improved information provision at the time of NBS [30] and preparation for and communication of positive NBS results may have improved [33]. This may be attributable to the increasing knowledge and experience of healthcare providers undertaking these roles and the growing body of evidence recognizing the potential negative outcomes should this be undertaken sub optimally.

However, other negative sequalae such as parental concerns about future reproductive decisions [17,20,22,29] and increased healthcare usage [21,25,26,27] did not seem to change over time, indicating that there is still more work to be done to ensure the harm associated with false-positive NBS results is mitigated.

Following communication of a positive NBS result, parents should be provided with written information (in person or via email depending on the mode of delivery) to ensure they have the name of the suspected condition and understand what will happen next. This should include links to trusted websites should families wish to learn more about the process or the condition [10,23,41]. This is important since despite parents frequently being advised not to use the internet to search for information about their child’s NBS result during the time between communication of the initial NBS result and confirmatory diagnostic texting, studies have found that almost all parents do this and that it can cause more harm than good [7,10,55]. Finally, minimizing the time between communication of the positive NBS result and confirmatory diagnostic testing [17,18] is another important measure to reduce potential anxiety in parents while awaiting the outcome of NBS.

## 5. Limitations

Ten of the included studies focused on cystic fibrosis, which may limit the transferability of the findings and recommendations. However, many of the strategies identified to potentially reduce the impact of false-positive results on families are generic and focus more on the stages of the NBS journey rather than condition-specific outcomes. As this was a scoping review, the search was limited to two databases. A systematic review may have identified more studies that could have further elucidated the psychosocial impact of false-positive NBS results. However, this provides a useful starting point to understand the potential psychosocial impact of false-positive NBS results and therefore possible areas for improvement and future intervention.

## 6. Conclusions

False-positive results following NBS have the potential to lead to negative psychosocial outcomes for families. Studies to date have not definitively identified factors that increase or decrease the risk; this does not appear to be condition- or process-dependent. Despite this, good practice points can be deduced from these studies which, if implemented, have the potential to manage parental expectations and reduce negative sequelae. These include providing information during the antenatal period and at the time of taking the NBS sample about potential false-positive outcomes; giving parents choices about how they would like to receive their child’s NBS result; ensuring the mother is not alone when the NBS result is communicated; tailoring information when delivering a positive NBS result so parents are aware it is a screening rather than a diagnostic result; and following up any verbal communication with written resources including signposting to reliable sources of further information and reducing the time between the NBS result being communicated and confirmatory diagnostic testing. Future research should focus on specific strategies to support parents at all stages of the screening journey for a range of outcomes.

## Figures and Tables

**Figure 1 children-11-00507-f001:**
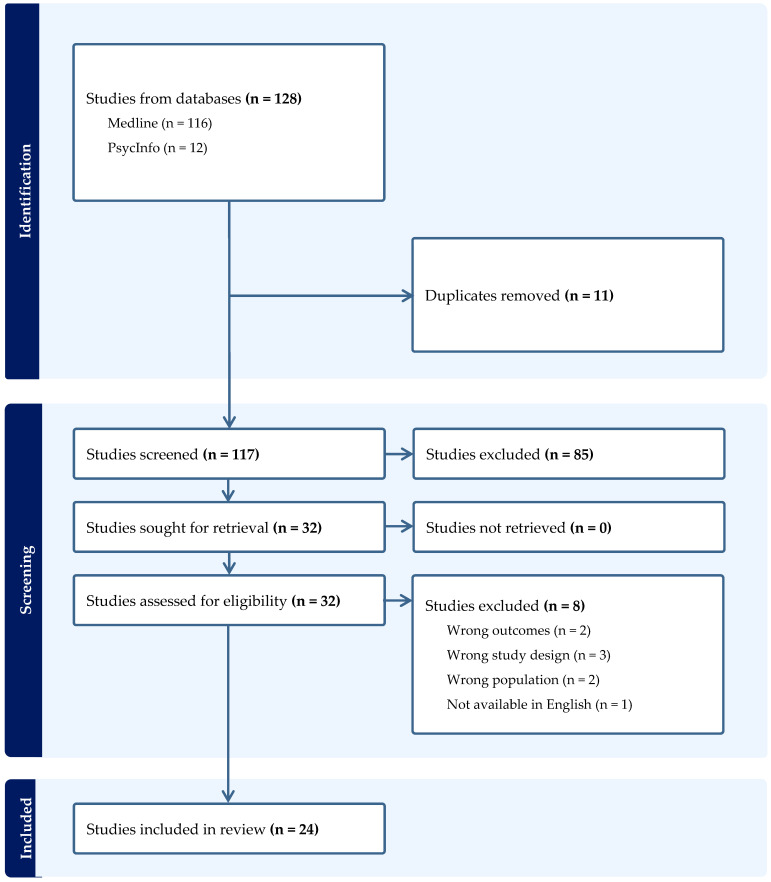
PRISMA flowchart of search results.

**Table 1 children-11-00507-t001:** Characteristics of included studies.

Author, Year, and Country	Condition(s)	Research Design/Data Collection Methods	Sample	Psychosocial Impact of False-Positive NBS Results
[15] Gurian, E.A.; Kinnamon, D.D.; Henry, J.J.; Waisbren, S.E. (2006) US	Biochemical genetic disorders	Mixed methods: structured interview; Parenting Stress Index	Parents of 173 children with false-positive NBS results and 67 children with normal NBS results.	Parents of children with a false-positive NBS result scored higher on the Parenting Stress Index.
[16] O’Connor, K.; Jukes, T.; Goobie, S.; DiRaimo, J.; Moran, G.; Potter, B.K.; Chakraborty, P.; Rupar, C.A.; Gannavarapu, S.; Prasad, C. (2018) Canada	Inborn error of metabolism, an endocrine disorder (congenital adrenal hyperplasia and congenital hypothyroidism) or cystic fibrosis (CF)	Cross-sectional study: Parenting Stress Index; Depression, Anxiety and Stress Scale	Mothers who had received a true-negative result (*n* = 31), true-positive result (*n* = 8), and a false-positive result (*n* = 18).	There were no significant differences regardless of NBS outcome in terms of overall anxiety, stress and depression.
[17] Hayeems, R.Z.; Miller, F.A.; Barg, C.J.; Bombard, Y.; Kerr, E.; Tam, K.; Carroll, J.C.; Potter, B.K.; Chakraborty, P.; Davies, C.; et al. (2016) Canada	CF	Mixed-methods cohort design: prospective self-report data; qualitative interviews	134 mothers of infants with false-positive CF NBS results completed questionnaires. 54 mothers of infants with false-positive CF NBS results were interviewed.	Mothers reported psychosocial distress but this was not detected via the psychosocial response measures at the newborn stage or 1 year later.
[18] Moran, J.; Quirk, K.; Duff, A.J.; Brownlee, K.G. (2007) UK	CF	Qualitative study: semi-structured interview	21 parents.	Mothers described experiencing anxiety, stress and upset during the screening process.
[19] Beucher, J.; Leray, E.; Deneuville, E.; Roblin, M.; Pin, I.; Bremont, F.; Turck, D.; Ginies, J.L.; Foucaud, P.; Rault, G.; et al. (2009) France	CF	Mixed Methods: interviews; Perceived Stress Scale; Vulnerable Child Scale	86 children with CF.	Mean Perceived Stress Scale Score did not differ when compared with the French population. Mean Vulnerable Child Scale was high. Results at 1 and 2 years did not differ significantly.
[20] Tluczek, A.; Orland, K.M.; Cavanagh, L. (2011) US	CF	Qualitative study: interviews	87 parents of 44 infants who had received false-positive NBS results for CF.	Repercussions of receiving genetic information following NBS included concern about test accuracy, the child’s health, questioning paternity, wondering if other relatives were carriers, and searching for the genetic source Positive outcomes included gaining new perspectives and strengthening relationships.
[21] Waisbren, S.E.; Albers, S.; Amato, S.; Ampola, M.; Brewster, T.G.; Demmer, L.; Eaton, R.B.; Greenstein, R.; Korson, M.; Larson, C.; et al. (2003) US	Biochemical genetic disorders	Mixed methods: children’s health and development; Parental Stress Index; interviews	Families of 50 affected children identified through expanded NBS; 33 affected children identified clinically, 94 children with false-positive results, 81 children with negative NBS results.	Children with a false-positive NBS result were twice as likely as children in the negative NBS result group to experience hospitalization. Mothers of children in the false-positive group obtained higher scores on the Parental Stress Index compared to mothers of children in the negative NBS group.
[22] Morrison, D.R.; Clayton, E.W. (2011) US	Metabolic or endocrine conditions included in NBS	Mixed methods: telephone interviews; Parental Stress Index-Short Form	Parents of children with false-positive but otherwise healthy NBS results (*n* = 28), true positive (*n* = 20) and false positive with other medical conditions (*n* = 12).	Parents who had received a false-positive NBS result expressed concern about having more children. Almost 10% of parents who had received a false-positive NBS result reported clinically significant stress as well as concern about their child’s health and future.
[23] Schmidt, J.L.; Castellanos-Brown, K.; Childress, S.; Bonhomme, N.; Oktay, J.S.; Terry, S.F.; Kyler, P.; Davidoff, A.; Greene, C. (2012) US	Any condition included in NBS	Qualitative study: semi-structured interviews; focus groups	27 parents of children who had received a false-negative NBS result.	Most parents did not report long-term negative impacts following a false positive NBS result but some experienced residual worry. Some parents identified positive outcomes after receiving the false positive NSB result.
[24] Lipstein, E.A.; Perrin, J.M.; Waisbren, S.E.; Prosser, L.A. (2009) US	Any of 20 biochemical disorders	Mixed methods: telephone survey; Parental Stress Index	200 children with false-positive NBS results and 137 children with negative NBS results.	There were no significant differences between NBS results and child healthcare utilization.
[25] Hayeems, R.Z.; Miller, F.A.; Vermeulen, M.; Potter, B.K.; Chakraborty, P.; Davies, C.; Carroll, J.C.; Ratjen, F.; Guttmann, A. (2017) Canada	CF	Population-based cohort study using linked health administrative data	1564 false-positive CF results and 6256 screen-negative matched controls.	A greater proportion of infants with false-positive results had more than 2 outpatient visits and more than 2 hospital admissions compared with control. No differences in healthcare use among mothers were detected.
[26] Tu, W.J.; He, J.; Chen, H.; Shi, X.D.; Li, Y. (2012) China	Metabolic disorders	Mixed methods: interviews; Parental Stress Index	Parents of 49 children with false-positive NBS results and 42 children with negative NBS results.	More mothers who had received a false-positive NBS result, worried about their child’s future compared to the negative NBS group. There were no differences in terms of worry for fathers in both groups. Children with false-positive NBS results were three times more likely to experience hospitalization.
[27] Karaceper, M.D.; Chakraborty, P.; Coyle, D.; Wilson, K.; Kronick, J.B.; Hawken, S.; Davies, C.; Brownell, M.; Dodds, L.; Feigenbaum, A.; et al. (2016) Canada	Medium-chain acyl-CoA dehydrogenase deficiency	Cohort study using healthcare administrative data sets	43 infants with a false-positive NBS result.	Children who had received a false positive NBS result experienced significantly higher physician visits (IRR 1.42) and hospitalizations (IR 2.32) during the first year compared with children who had received a negative (normal) NBS result.
[28] Hewlett, J.; Waisbren, S.E. (2006) US	CF Metabolic disorders Congenital hypothyroidism Newborn hearing screening	Literature review: using Medline (via PubMed) and Journals@OVID	Nine studies were included.	Improved education and communication with parents, specifically at the time of follow up screening, can reduce parental stress and anxiety associated with a false-positive NBS result.
[29] Tluczek, A.; Mischler, E.H.; Bowers, B.; Peterson, N.M.; Morris, M.E.; Farrell, P.M.; Bruns, W.T.; Colby, H.; McCarthy, C.; Fost, N.; et al. (1991) US	CF	Qualitative: semi-structured interviews	Parents of infants with a false-positive result (*n* = 104); control group (*n* = 11).	Communication via telephone was more likely to lead to misunderstanding of the sweat test result than in-person communcation. Some parents expressed ongoing concerns that their child may still have CF. Most parents did not change their reproductive plans. Parents in both groups experienced strong emotional responses. These included worry but also gratitude in terms of their child being identified early if they had of had a problem.
[30] Gapp, S.; Garbade, S.F.; Feyh, P.; Brockow, I.; Nennstiel, U.; Hoffman, G.F. Sommerburg, O.; Gramer G, G.; 2022 Germany	CF	Cross-sectional: prospective questionnaire-based survey	178 families responded to the emainedaire (33.7% had a confirmed CF diagnosis).	17% of families with a false-positive NBS result remained anxious following confirmatory diagnostics.
[31] Lang, C.W.; McColley S.A.; Lester, L.A.; Friedman Ross, L. 2011 US	CF	Cross-sectional: telephone survey	90 parents.	94% of parents understood their child did not have CF after a negative sweat test. 79% of those identified as carriers understood their result. Parents expressed frustration related to the lack of knowledge and sensitivity by those who notified them of the initial positive NSB result. Speaking to a genetic counsellor while waiting for the sweat test decreased anxiety.
[32] Green, J.M.; Hewison J.; Bekker, H.l.; Bryant, L.D.; Cuckle, H.S. 2004 UK	All screened conditions	Systematic review	106 papers were included: 78 about antenatal screening and 28 about NBS.	Anxiety in parents who had received false positive NBS resultmay not return to normal levels. Residual anxiety may continue, possibly over extended periods of time.
[33] Peterson, L.; Siemon, A.; Olewiler, L.; McBride, K.L.;Allain, D.D.; 2021 US	Krabbe Disease	Qualitative: cross-sectional semistructured, audio-recorded, phone interviews	11 families were included in the analysis.	Parents reported experiencing emotional distress and negative emotions, specifically uncertainty during the initial communication of the NBS result, and leading up to and during the follow-up appointment with a genetic counsellor.
[34] Tluczek, A.; Mischler E.H.; Farrell P.M.; Fost, N.; Peterson, N.M.; Carey, P.; Bruns, W.T.; McCarthy, C. 1992 US	CF	Cross-sectional: survey	Parents of 104 infants with a false-positive NBS result for CF	Parents expressed shock, disbelief and confusion in response to the initial positive NBS result. Most parents (92%) were relieved following the negative sweat test and (88%) were aware that a negative sweat test meant their child did not have CF. Communication via telephone was more likely to lead to misunderstanding of the sweat test results compared to those told in person. Six parents who had lingering concerns that their child might have CF were informed of the sweat test via the telephone. The children also had lower APGAR scores at birth. 69% did not change their reproductive plans as a result of the false-positive result, 8% definitely changed their plans and 22% were uncertain.
[35] van den Heuvel, L.M.; van der Pal, S.M.; Verschoof-Puite, R.K.; Klapwijk, J.E.; Elsinghorst, E.; Dekker, E.; van der Ploeg C.P.B.; Henneman, L. 2024 The Netherlands	All screened conditions	Cross-Sectional: questionnaire; Hospital Anxiety and Depression Scale; Child Vulnerability Scale; TNO-AZL Preschool Children Quality of Life Scale (TAPQoL); NL-TiC-P questionnaire	112 parents completed the questionnaires, of whom 35 had received a true-positive result, 20 a false-positive, 57 an inconclusive result and 268 controls 5 weeks after NBS. 60 parents completed the questionnaires, of whom 19 had received a true-positive result, 14 a false-positive, 27 an inconclusive result and 116 controls 4 months after NBS.	Negative emotions were more commonly associated with parents who had received true-positive, false-positive and inconclusive results both following the initial NBS result and four months later. No significant differences were observed in parental perceptions of child vulnerability between true-positive, false-positive, inconclusive and control groups after receiving the initial NBS result and four months later. Parents of children born at term who received true-positive and false-positive NBS results reported more frequent visits to a paediatrician compared with controls both after receiving the initial NBS result and four months later.
[36] Vernooij-van Langen, A.M.M.; van der Pal, S.M.; Reijntjens, A.J.T.; Loeber, J.G.; Dompeling, E.; Dankert, Roelse, J.E. 2014 The Netherlands	CF	Prospective controlled study using mixed methods: semi-structured interviews; questionnaires; Hospital Anxiety and Depression Scale; TNO-AZL Preschool Children Quality of Life Scale (TAPQoL)	Questionnaires were received from 62 parents who received a false-positive result, and 146 controls. Twenty-four mothers and three fathers took part in 25 interviews.	Parents in both the false-positive and control groups considered their child to be low risk in terms of being affected by CF. Parents in the false-positive group considered CF NBS less reliable. Scores related to anxiety and depression were not significantly different between the false-positive and control groups. Parental concern about their child’s health and number of GP visits did not differ between groups.
[37] Sorenson, J.R.; Levy, H.L.; Mangione, T.W.; Sepe, S.J. 1984 US	Metabolic conditions	Mixed methods: telephone interviews; parental psychological status was assessed using the Multipled Affect Adjective Checklist	60 parents.	Parents who received an abnormal NBS result were no more anxious or depressed while waiting for the repeat test results than other parents. Parents were less anxious and depressed after the repeat test was found to be ‘normal’. However, 36% of parents reported concern about the health of their infant because of the need for the repeat test.
[38] Tarini, B.A.; Clark, S.J.; Pilli, S.; Dombkowski, K.J.; Korzeniewski, S.j.; Gebremariam, A.; Eisenhandler, J.; Grigorescu, V. 2011 US	All screened conditions	Cohort study using medicaid data	818 infants with a false-positive NBS result.	Pre-term but not term infants with false-positive NBS results had more acute outpatient visits than infants who had received normal NBS results.

## Data Availability

No new data were created or analysed in this study. Data sharing is not applicable to this article.

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
