# Peer review of "Psychosocial Impact of False-Positive Newborn Screening Results: A Scoping Review"

_children, 2024, doi:10.3390/children11050507_

Round 1

Reviewer 1 Report

Comments and Suggestions for Authors

The authors stated using the PRISMA checklist and there is  no evidence of having used this checklist.

Several methodological procedures (Methods section) are not reported such as not describing search terms, methods used to abstract or synthesize results, exploring reasons for heterogeneity.

It is difficult to assess the results because the methods are not  reported.

Comments on the Quality of English Language

In the abstract needs proof reading "...  potential to cause lead to significant harms..".

The preposition "for" is dangling after the word 'screened". The authors stated: "While the number of conditions screened for differ between countries..." should be "While the number of conditions screened differ between countries.."

The authors do not seem to have an understanding of the current medical model which includes screening. See line 39 onward.

Author Response

Thank you so much for your helpful comments and feedback.

The authors stated using the PRISMA checklist and there is  no evidence of having used this checklist.

The PRISMA flowchart has now been included. Apologies, I submitted this as the graphic to accompany the paper but have subsequently realized this was incorrect. 

Several methodological procedures (Methods section) are not reported such as not describing search terms, methods used to abstract or synthesize results, exploring reasons for heterogeneity.

Apologies, I think I was overzealous in my attempt to make the paper pithy and in so doing did not include enough information in the methods section. This has not been rectified with an updated and improved description of the search. 

It is difficult to assess the results because the methods are not  reported.

Apologies, this has now been rectified as per the above comment. In addition, a table of characteristics of included studies has been added to the results. 

In the abstract needs proof reading "...  potential to cause lead to significant harms..".

Apologies, this was a cut and paste error. This has been corrected and the whole paper has been proof read. 

The preposition "for" is dangling after the word 'screened". The authors stated: "While the number of conditions screened for differ between countries..." should be "While the number of conditions screened differ between countries.."

Thank you, this has been amended. 

The authors do not seem to have an understanding of the current medical model which includes screening. See line 39 onward.

Apologies, in hindsight (a wonderful thing!) it was clear this was not the message we were trying to convey. This has been amended. 

Thank you once again for your helpful suggestions.

Reviewer 2 Report

Comments and Suggestions for Authors

This article deals with a very important topic of newborn screening, as false positive results can reduce the acceptance of screening in addition to the sequelae for the affected parents. The article examines the psychosocial consequences of such NBS results based on a scoping review and describes strategies to reduce parental stress

In my opinion, however, the methodology of the article is inadequate. Even if it is "only" a scoping review, the search strategy and the exclusion criteria should be described. When entering ((((((newborns) AND (screening)) AND (positive)) AND (false)) AND (parents)) NOT (heart)) NOT (hearing)) in MEDLINE, for example, 1748 results are displayed for the years 2013-2023, 57 after 2018 alone. However, the last paper cited in the article from the database search is from 2018. In addition, it is actually recommended to search at least 2 databases. Therefore, that only 40 papers were found in the search and no further studies were identified via the reference list seems somewhat unlikely. At the very least, it should be stated how the 40 papers were selected and perhaps the search should be updated.

Otherwise, I only have minor issues for the discussion:

Perhaps one could also discuss what the content of the information about abnormal screening results should be and that the staff providing the information should be well trained in terms of content and communication to reduce parental distress about a (false) positive screening result. There is also literature on the advantages and disadvantages of providing information through a specialist or a known GP. The importance of an optimal screening algorithm with the best possible PPV should be emphasized. For cystic fibrosis, for example, this is not optimal in most countries, so that there are many false-positive findings, which explains the frequency of the literature on this subject.

Line 193: please add a reference (i. e. WHO Screening programmes 2021, and/or Andermann 2008)

Line 212-213. End of sentence is missing?!

Line 224: Please add the main strategies to reduce negative sequelae here again.

Author Response

Thank you so much for your helpful comments and feedback.

This article deals with a very important topic of newborn screening, as false positive results can reduce the acceptance of screening in addition to the sequelae for the affected parents. The article examines the psychosocial consequences of such NBS results based on a scoping review and describes strategies to reduce parental stress

In my opinion, however, the methodology of the article is inadequate. Even if it is "only" a scoping review, the search strategy and the exclusion criteria should be described. When entering ((((((newborns) AND (screening)) AND (positive)) AND (false)) AND (parents)) NOT (heart)) NOT (hearing)) in MEDLINE, for example, 1748 results are displayed for the years 2013-2023, 57 after 2018 alone. However, the last paper cited in the article from the database search is from 2018. In addition, it is actually recommended to search at least 2 databases. Therefore, that only 40 papers were found in the search and no further studies were identified via the reference list seems somewhat unlikely. At the very least, it should be stated how the 40 papers were selected and perhaps the search should be updated.

Apologies, I think I was overzealous in my attempt to make the paper pithy and in so doing did not include enough information in the methods section. This has not been rectified with an updated and improved search. 

Otherwise, I only have minor issues for the discussion:

Perhaps one could also discuss what the content of the information about abnormal screening results should be and that the staff providing the information should be well trained in terms of content and communication to reduce parental distress about a (false) positive screening result. There is also literature on the advantages and disadvantages of providing information through a specialist or a known GP. The importance of an optimal screening algorithm with the best possible PPV should be emphasized. For cystic fibrosis, for example, this is not optimal in most countries, so that there are many false-positive findings, which explains the frequency of the literature on this subject.

Thank you for this suggestion. We have revisited the discussion and provided ore in depth consideration of the findings, particularly in relation to who is communicating the results and how this is done. 

Line 193: please add a reference (i. e. WHO Screening programmes 2021, and/or Andermann 2008)

Thank you for this helpful suggestion, we have added the WHO reference. 

Line 212-213. End of sentence is missing?!

Apologies, this was a cut and paste error that has now been rectified. 

Line 224: Please add the main strategies to reduce negative sequelae here again.

Thank you for this helpful suggestion. These have now also been added to the conclusion. 

Round 2

Reviewer 1 Report

Comments and Suggestions for Authors

The authors are commended for conducting a scoping review on screening newborns which is a very important topic. The authors have made some improvements; however, additional attention is needed in a number of areas.

Abstract :

It is indicated that 16 papers were included; however, 15 papers are noted in the flow diagram, table and results.

False positives may affect coping capacities but causing or leading to significant harms seems like an overstatement and not supported by findings in this review.

It is stated that findings are inconsistent, and this statement might not be completely accurate. The authors may be seeing different experiences by different samples. 

Introduction

3rd paragraph, line 39: The introductory sentence for this  paragraph is inconsistent with the rest of the paragraph. This paragraph appears to be focused on the complexities of the process and situation and not on notions of health and illness.

Methods

The search terms appear inadequate for conducting a full or broad search. It seems many relevant studies may not have been located or included (eg  Peterson, Laiken et al. “A qualitative assessment of parental experiences with false-positive newborn screening for Krabbe disease.” Journal of genetic counseling vol. 31,1 (2022): 252-260. doi:10.1002/jgc4.1480). 

Perhaps consult with a librarian to expand search.

One inclusion criterion is stated as primary  research, though in contrast to this stated inclusion criterion, the exclusion criteria included case studies, which are primary research. 

What data were extracted? 

How were the themes created?

Flow diagram - why were 19 studies excluded (n=45 - n=26)

Table 1, what research designs were employed

Discussion

This statement requires verification (line 257)  " A notable finding of the present review was that increased stress and anxiety associated with a false positive NBS result was more commonly reported in earlier studies 258 [14,20,21,27,28]. This contrasts with some of the more recent studies which reported similar anxiety and stress levels in parents who had received false positive NBS results compared to parents who had received negative NBS results [15,16]. This is promising and 261 suggests that preparation for and communication of positive NBS results may have im-262 proved in recent years."

Many limitations should be  added to this section such as the shortcomings in the search and reporting. 

Comments on the Quality of English Language

Quality of writing

Delete "for" after the word, screened, most times.    

Author Response

Dear Reviewer,

Thank you for looking at this paper again. Please find responses below:

The authors are commended for conducting a scoping review on screening newborns which is a very important topic. The authors have made some improvements; however, additional attention is needed in a number of areas.

Response: Thank you for your positive feedback

Abstract :

It is indicated that 16 papers were included; however, 15 papers are noted in the flow diagram, table and results.

Response: This has been corrected and updated in response to comments about the search below. 

False positives may affect coping capacities but causing or leading to significant harms seems like an overstatement and not supported by findings in this review.

Response: This has been changed to "...have the potential to result in negative sequelae..." in response to this comment. 

It is stated that findings are inconsistent, and this statement might not be completely accurate. The authors may be seeing different experiences by different samples. 

Response: Than you for pointing this out. This has been changed to 'variable' to acknowledge the different experiences. 

Introduction

3rd paragraph, line 39: The introductory sentence for this  paragraph is inconsistent with the rest of the paragraph. This paragraph appears to be focused on the complexities of the process and situation and not on notions of health and illness.

Response: This has been changed to 'parental perceptions' to make this clearer. 

Methods

The search terms appear inadequate for conducting a full or broad search. It seems many relevant studies may not have been located or included (eg  Peterson, Laiken et al. “A qualitative assessment of parental experiences with false-positive newborn screening for Krabbe disease.” Journal of genetic counseling vol. 31,1 (2022): 252-260. doi:10.1002/jgc4.1480). 

Perhaps consult with a librarian to expand search.

Response: Thank you so much for raising this. I too was confused about why this had not been picked up by the search. We did consult with a librarian initially and decided a sensitive rather than specific approach might be preferable to ensure we captured all papers about false positive NBS results. Following your observations, we sought additional advice and support from the librarians.

The librarians combined existing search terms (i.e., false positive, newborn screening and psychosocial) with the title of the paper above to determine why we were missing this paper (and others) using the existing search terms. It became apparent that our initial strategy (to use a sensitive approach) actually meant we were missing papers and in effect needed to use more specific terms to capture more papers. This led us to incorporating additional search terms to ensure we did not miss papers such as the one above and actually, in so doing we identified a further 9 paper for inclusion in the review.

I am hugely grateful to you for identifying this as it is something I was struggling to get to the bottom of and in fact the librarians are going to use this example in their teaching session to explain alternative ways of searching but also ensuring pertinent papers are not being missed and if so, how by a process of elimination, it is possible to determine which 'concept' may be the cause and how this can be rectified. 

This led to us running a completely new search using the additional search terms (these new search terms have been added to the methods). We uploaded the results into Covidence and redid the screening process and as mentioned included a further 9 papers which also led to us restructuring our results. We believe the paper has been greatly improved as a result, thank you once again. I was also so grateful to get to the bottom of this as it was really quite irksome! 

One inclusion criterion is stated as primary  research, though in contrast to this stated inclusion criterion, the exclusion criteria included case studies, which are primary research. 

Response: Apologies, this was an oversight. 

What data were extracted? 

Response: This has been described in more detail in the methods section. 

How were the themes created?

Response: This has been described in more detail in the methods section. 

Flow diagram - why were 19 studies excluded (n=45 - n=26)

Response: The flowchart is created in Covidence during the screening process and only includes reasons for exclusion at the full text stage. 

Table 1, what research designs were employed

Response: The research designs have been added as well as the additional included studies. 

We have also restructured the results section as a result of the identification of the additional 9 papers for inclusion. 

Discussion

This statement requires verification (line 257)  " A notable finding of the present review was that increased stress and anxiety associated with a false positive NBS result was more commonly reported in earlier studies 258 [14,20,21,27,28]. This contrasts with some of the more recent studies which reported similar anxiety and stress levels in parents who had received false positive NBS results compared to parents who had received negative NBS results [15,16]. This is promising and 261 suggests that preparation for and communication of positive NBS results may have im-262 proved in recent years."

Response: We have removed the beginning of this sentence to make this clearer. 

Many limitations should be  added to this section such as the shortcomings in the search and reporting. 

Response: We have added additional limitations to the limitations paragraph. 

Thank you once again for your suggestions about the search as this really did help to identify what the issue was with the existing search terms.

With best wishes

Jane

Reviewer 2 Report

Comments and Suggestions for Authors

I already have reviewed this paper!

Author Response

Thank you for reviewing this paper, an updated version in response to the comments from Reviewer 1 is attached.
